# MicroRNA (miR)-124: A Promising Therapeutic Gateway for Oncology

**DOI:** 10.3390/biology12070922

**Published:** 2023-06-28

**Authors:** Karthik Gourishetti, Vignesh Balaji Easwaran, Youssef Mostakim, K. Sreedhara Ranganath Pai, Deepak Bhere

**Affiliations:** 1Biotherapeutics Laboratory, School of Medicine Columbia, University of South Carolina, Columbia, SC 29209, USA; 2Department of Pathology, Microbiology, and Immunology, School of Medicine Columbia, University of South Carolina, Columbia, SC 29209, USA; 3Department of Pharmacology, Manipal College of Pharmaceutical Sciences, Manipal Academy of Higher Education, Manipal 576104, India; 4College of Arts and Sciences, University of South Carolina, Columbia, SC 29208, USA

**Keywords:** miR-124, oncology, microRNA therapeutics, cancer treatment, clinical implications

## Abstract

**Simple Summary:**

MicroRNA-124 (miR-124) is a small non-coding RNA that regulates gene expression and is abundantly expressed in the brain and immune system. Dysregulated expression of miR-124 is associated with several cancer types, making it a potential therapeutic target in oncology. We demonstrate the potential of miR-124 as a target in various cancer types.

**Abstract:**

MicroRNA (miR) are a class of small non-coding RNA that are involved in post-transcriptional gene regulation. Altered expression of miR has been associated with several pathological conditions. MicroRNA-124 (miR-124) is an abundantly expressed miR in the brain as well as the thymus, lymph nodes, bone marrow, and peripheral blood mono-nuclear cells. It plays a key role in the regulation of the host immune system. Emerging studies show that dysregulated expression of miR-124 is a hallmark in several cancer types and it has been attributed to the progression of these malignancies. In this review, we present a comprehensive summary of the role of miR-124 as a promising therapeutic gateway in oncology.

## 1. Introduction

MicroRNA (miR) are a group of non-coding RNA containing ~25 nucleotides that have significant roles in gene regulation. miR play a pivotal role in modulation of gene expression in various biological processes such as proliferation, differentiation and maturation of cells, migration, metastasis, cell death, and autophagy [1].

miR-124 was discovered in the year 2002 [2], and to date three subtypes of human miR-124 have been identified. They are miR-124a-1(8p23.1), miR-124a-2(8q12.3), and miR-124a-3 (20q13.33) and are abundantly expressed in neuronal cells [3]. Like all miR, biogenesis of miR-124 takes place in the cytoplasm and nucleus of the cell (Figure 1). In the nucleus, genes of miR-124 are produced through the RNA polymerase II (RNA Pol II) to synthesize a stem-loop structure called pri-miR-124, which is about 100–120 nucleotides. Pri-miR-124 then gets converted into pre-miR (70–100 nucleotides) with the help of DROSHA, DiGeorge syndrome chromosomal region 8 (DGCR8), and RNase III. Pre-miR-124 is then transported to the cytoplasm by exportin 5. In the cytoplasm, pre-miR-124 produces the miR-124 duplex. This conversion is facilitated by DICER and transactivation response element RNA-binding protein (TRBP) with RNase III. DICER then breaks down the hairpin structure of the pre-miR-124 and induces double-strand miR-124 formation without the hairpin. DICER is an endonuclease with double RNase III domains supported by TRBP to cleave the miR precursor and produce mature miR. The mature miR is then integrated into the RNA-induced silencing complex (RISC), loaded with an argonaut protein and then bound to target mRNA along with the complementary sequences to stimulate the mRNA strand degradation and translation inhibition [4].

miRs possess a dual function, as they not only play a cytoplasmic role but can also be translocated into the nucleus, where they regulate gene expression in various ways [5]. For instance, miRs can bind to long non-coding RNAs (lncRNAs) and efficiently inhibit their function. As a matter of fact, lncRNAs are transcripts that do not encode proteins but can regulate gene expression [6]. Furthermore, miRs can attach to transcription factors, which are proteins that control the expression of other genes. Although the exact mechanism of miR translocation into the nucleus is not entirely known, it is widely believed that the importin 8 (IPO8) protein plays a pivotal role in this process. IPO8, being a nuclear import receptor, recognizes and binds to a nuclear localization signal (NLS) present on the miRNA-containing complex. Consequently, the complex is transported into the nucleus with the help of the nuclear pore complex [7]. By targeting lncRNAs, transcription factors, and other nuclear transcripts, the translocation of miRNAs into the nucleus offers an additional level of regulation for these vital molecules, which can have a profound impact on gene expression at the genomic level.

## 2. Biological Functions of miR-124

miR-124 is one of the most abundantly expressed miR in the central nervous system (CNS). It regulates several key biological processes, such as cell proliferation, neuronal differentiation, autophagy, and inflammation [8]. miR-124 also plays a significant role in suppressing the oncogenic modification of normal cells in other tissues, like breast and pancreas [9,10,11,12,13]. Several studies have reported that down-regulation of miR-124 leads to the development of advanced cancers in the brain, breast, and colon [14,15].

miR-124 plays pivotal roles in the regulation of metabolism [16]. Reports have demonstrated that among several miR, miR-124 expression is abundant in the liver and regulates homeostasis of fatty acids and triglycerides. Genome-wide expression analysis revealed that miR-124 down-regulates the genes which are involved in fatty acid oxidation (β-oxidation) and triglyceride hydrolysis [17].

## 3. Role of miR-124 in Cancer 

### 3.1. Neurological Cancers

Glioblastoma (GBM) is an aggressive brain tumor (WHO Grade IV astrocytoma) associated with a poor survival rate [18]. 

Polypyrimidine tract-binding protein (*PTBP*) *1* is an oncoprotein that supports the growth of tumor cells and maintains the metastatic potential of cancer. *PTBP1* amplification was found in GBM due to the loss of brain-enriched miR-124 [19]. miR-124 was significantly reduced in GBM and exogenous transfection of miR-124 into GBM cells induced a G_0_/G_1_ cell cycle phase arrest and reduced the expression of cyclin-dependent kinase (*CDK*)*6* and phosphorylated retinoblastoma (pRb) proteins [20]. miR-124 targets many proteins like *SCP1*, Rho-associated protein kinase 1 (*ROCK1*), *STAT3*, matrix metallopeptidase 9 (*MMP9*), and inhibits tumor cell proliferation in GBM [4]. miR-124 targets cyclin-dependent kinase-4 (*CDK4*) and sensitizes cells to radiotherapy. Exogenous delivery of miR-124 enhances temozolomide sensitivity in GBM cells, such as U87MG. In addition, it reduces the migration of tumors by targeting *CDK6* [21] GBM cells like U87MG and T98G, which have a high expression of the clock circadian regulator (*CLOCK*) gene; this plays a vital role in maintaining tumorigenesis. miR-124 can effectively silence the *CLOCK* gene directly, by inhibiting the activation of NF-kB. Overexpression of miR-124 decreases *SOX9* protein, reduces tumorigenicity, and enhances radiosensitivity of GBM cells [22]. miR-124 has been found to be a potent anti-glioma molecule against glioma stem cells (GSC) [23]. Exosome delivery (exo) with miR-124a (exo-miR-124a) revealed a significant decrease in the viability and clonogenicity in the intracranial GSC xenograft model compared to a control. miR-124a targets and downregulates Forkhead box A2 (*FOX A2*), an oncogenic transcription factor, and intermediary of lipid metabolism in GSCs. In vivo studies revealed that exo-miR-124 was found in around 50% of animals living long term when compared to a control, suggesting that exo-miR-124 is an effective anti-glioma agent [24]. miR-124 targets aurora kinase A (*AURKA*), inhibits growth of LN229 GBM cells, and potentiates chemosensitivity against temozolomide [25]. miR-124 targets *NRAS*, *PIM3*, and inhibits GSC proliferation and growth [26]. Neuropilin-1 (*NRP-1*) receptors are expressed in various cancers including glioma cells. miR-124 particularly targets NRP-1, as well as PI3K/Akt/NFkB signaling pathway, which inhibits tumor progression [27]. Upregulation of p62 oncogene is targeted and inhibited by miR-124-3p and serves as a novel therapeutic molecule to control glioma cell progression [28]. miR-124 directly interacts and inhibits the signal transducer and activator of transcription 3 (*STAT3*) signaling pathway and acts as an immuno-therapeutic molecule in the GSC tumor microenvironment. Upregulation of miR-124 acts as a potent anti-tumor agent and inhibits GBM cell invasion [29]. Syndecan binding protein (*SDCBP*) was widely distributed in intracellular proteins containing physiological and pathological role in cancers. miR-124-3p upregulation depletes the *SDCBP* expression and inhibits the proliferation, migration, and invasion of GBM [29]. 

Pilocytic astrocytoma, a common pediatric cancer type, is associated with a high mortality rate and poor prognosis. Irregular expression of miR-124 was identified in pilocytic astrocytoma tissues compared to healthy brain tissues. There is a proven correlation between miR-124 downregulation and pilocytic astrocytoma [30]. miR-124-3p has emerged as a potential biomarker and an effective therapeutic molecule for the treatment of ependymomas by targeting tumor protein p53 nuclear protein 1 (*TP53INP1*) [31]. In medulloblastoma, miR-124 was expressed 6.5-fold lower than in normal cerebellum [4]. miR-124 acts as a tumor suppressor gene in medulloblastoma pathogenesis by inhibiting cell cycle progression in the G_0_/G_1_ phase without affecting apoptosis, slows down tumor growth by targeting *CDK6* proto-oncogene, and inhibits cell proliferation in DAOY and D283 cells and solute carrier family 16 (*SLC16A1*) [32]. Several transcriptional factors such as *SOX9*, Forkhead box protein G1 (*FOXG1*), and *MEIS1* are regulated by miR-124 in medulloblastoma [33]. The nuclear receptor *Nur77* is upregulated in medulloblastoma, thus acting as an oncogene promoter by inducing cell proliferation and tumor spheroid size. Modulating miR-124 to physiological levels suppressed the *Nur77* and prevented cancer progression [34]. miR-124 thereby is considered a promising therapeutic miR in medulloblastoma patients with elevated *Nur77* protein. 

miR-124 is highly enriched in primary CNS lymphoma [35]. miR-124 was aberrantly expressed in the pituitary adenoma [36]. miR-124 suppressed the migration and invasion of pituitary adenoma cells by targeting *FSCN1*, pituitary tumor-transforming gene 1 protein-interacting protein (*PTTG1IP*), and Ezrin (*EZR*) [37]. Targeting miR-124 could potentially play a key role in various neurological cancers including GBM.

### 3.2. Breast Cancer

Breast Cancer (BC) is a very prevalent in women and is associated with high morbidity and mortality rates. BC has a high metastatic potential affecting physical and psychological health. The expression of miR-124 has been found to be low in BC tissues, thereby promoting invasion and metastatic potential [38]. Transfection of miR-124 into MDA-MB-231 cells suppressed tumor cell progression and increased the sensitivity to chemotherapy [39]. The restoration of miR-124 to physiological levels improved survival outcomes in breast invasive carcinoma as compared with miR-124 low mammary carcinoma [40]. miR-124 suppressed the BC cells induced by bone metastasis via downregulating IL-11 [39]. *CD151* is a tetraspanin family member that regulates cell development, growth, and motility. *CD151* is over expressed in the BC cell lines MCF-7 and MDA-MB-231. miR-124 targets *CD151* and inhibits rapid proliferation, suggesting that *CD151* is a potential mediator for miR-124 mediated targeting of BC [41]. 

Studies revealed that *CDK4* was a target for miR-124 in the MCF-7 cell line [42]. Restoration of miR-124 levels were found to lower cell proliferation, viability, and growth of BC [42]. Cell cycle was arrested in the G_1_/S phase by miR-124 via *EGFR* signaling that further impedes cell proliferation in BC [43]. *AKT2* is a potential biomarker and is targeted by miR-124 in ERα-positive BC cells [41]. Ets-1 is an oncoprotein that regulates tumor progression and survival in BC. *ETS-1* is a potential target for miR-124 and controls the growth of BC cells [44]. Flotillin-1 (*FLOT1*) is a novel target for miR-124. miR-124 is considered to be a tumor suppressor protein via regulation of *FLOT1* protein. *FLOT1* is overexpressed in BC and is inhibited by miR-124 and control cell proliferation and migration [39]. Doxorubicin resistance was reversed by miR-124 via the STAT3/HIF-1 signaling pathway [39]. miR-124 level was downregulated in BT474, SKBR3, and MCF7 via monocarboxylate transporter 1 (*MCT1*) (*SLC16A1*)-mediated glucose metabolism in BC cell lines. Transfection of miR-124 into these cells depletes Taxol-resistance and aids the killing of cancer cells [45]. These studies demonstrate the potential role of miR-124 to target breast cancer.

### 3.3. Hepatocellular Carcinoma (HCC) 

Hepatocellular carcinoma (HCC) is a liver tumor which accounts for over 90% of primary liver tumors. Hepatocellular carcinoma occurs in approximately 85% of patients diagnosed with cirrhosis [46]. Hepatocellular cancer (HCC) is another prevalent cancer with a high mortality rate. HCC is now the fifth most widespread form of cancer worldwide. Additionally, among men, it is the second leading cause of cancer-related deaths, behind lung cancer [47]. Previous studies have demonstrated that expression of miR-124-3p is lower in hepatocellular carcinoma (HCC) as compared to that of healthy hepatocytes [48]. 

Sorafenib is a multi-kinase inhibitor frequently used to treat patients with advanced hepatocellular carcinoma (HCC). Resistance to sorafenib is a significant hindrance to the treatment’s efficacy. miRNA-124-3p.1 has been found to sensitize HCC cells to sorafenib-induced apoptosis through the regulation of *FOXO3a* phosphorylation as well as deacetylation by targeting *AKT2* and *SIRT1* [49]. Aquaporin 3 (*AQP3*) is a type of aquaporin located in the plasma membrane of cells. In HCC, *AQP3* is often overexpressed, leading to the promotion of stem cell-like properties in hepatoma cells by regulating *CD133*. Additionally, AQP3 is a direct target of miR-124 and its expression can be suppressed through enrichment of miR-124 [50]. 

*Sp1* protein and *integrin αV* were directly targeted by miR-124, which plays a role in cell migration and invasion; this is demonstrated in SMMC-7721 and BEL-7404 cells [51]. HepG2 cell proliferation was inhibited via miR-124 transfection by targeting *STAT3* protein. Overexpression of miR-124 initiated HCC apoptosis [52]. miR-124 restoration in HCC leads to cell cycle arrest in G_1_ phase, affects cell proliferation, and reduces tumorigenesis [52]. In HCC cells, miR-124 reduces the activity of cancer susceptibility candidate 3 (*CASC3*) and deactivates key molecules involved in cell proliferation, such as extracellular signal-regulated kinase (*ERK*), *MAPK*, *p38*, phosphoinositide 3-kinase catalytic subunit alpha (*PIK3CA*), and *CD151*. miR-124-3p expression is associated with changes tumor size and its potential as a biomarker has been demonstrated in [53]. Studies have indicated that patients with CD133+ HCC are resistant to chemotherapy. By restoring miR-124, CD133+ HCC were responsive to cisplatin therapy, leading to apoptosis through the targeting of the SIRT1/ROS/JNK pathway [54]. miR-124 inhibited tumorigenesis and progression in HCC by targeting Kuppel-like factor 4 (*KLF4*) [55]. 

### 3.4. Lung Cancer

More than 85% of lung cancers (LC) have been identified as non-small-cell lung cancer (NSCLC). In NSCLC, miR-124-3p significantly suppressed metastasis through extracellular exosome transport and intracellular PI3K/AKT signaling [56]. miR-124 targets disintegrin and a metalloproteinase 15 (*ADAM 15*), which are related to several cellular regulations, including metastatic progression. miR-124 inhibits *ADAM15* and prevents NSCLC invasion. NSCLC cell invasion and migration are actively inhibited by miR-124 via repressing zinc finger E-box binding homeobox 1 (*ZEB1*) [57]. Overexpression of miR-124 inhibits *SOX9* and controls cell proliferation and migration in lung adenocarcinoma [58]. miR-124 acts as a tumor suppressor and inhibits cancer cell proliferation by targeting oncogenic *CD164* and Cadherin-2 (*CDH2*) signaling pathways in NSCLC [59]. Gefitinib resistance is a threat for NSCLC patients and low expression of miR-124 has been linked to gefitinib resistance in NSCLC patients. Upregulation of miR-124 in NSCLC inhibits *SNAI2* and *STAT3* and reverses gefitinib resistance in NSCLC, thus acting as a prognostic factor [60]. miR-124 controls the cellular glycolysis and metabolism processes via targeting *AKT1/2–glucose transporter 1/hexokinase II* in NSCLC [61]. *Rab27A* gets targeted by miR-124a and inhibits lung cancer cell lines like PC9 and H1299 [62]. *HOXA11-AS* expression was upregulated in A549 lung cancer cells and contributes to tumor size enlargement and lymph node metastasis. miR-124 reverses the *HOXA11-AS* expression in the lung cancer cell and halts tumor progression [63]. 

miR-124 disturbs autophagy and reduces cell survival by depleting p62, which is an autophagy regulator of the transcription factor *NF-kB* in the *KRAS* mutant NSCLC patients [64]. miR-124 represses autophagy by targeting sirtuins 1 (*SIRT1*) and improves the cisplatin sensitivity against NSCLC [65]. LIM-homeobox domain 2 (*LHX2*) plays an essential role in cell proliferation and differentiation. The aberrant nature of *LHX2* has been associated with cancer and promotes irregular cell proliferation. In NSCLC, *LHX2* is upregulated which in turn promotes cell growth in A549 and H1299 lung cancer cells. miR-124 represses the *LHX2* expression and inhibits migration, invasion, and arrests the cell cycle at the G1 phase in NSCLC [66]. *MYO10* expression is inhibited by miR-124 by regulating *NF-kB* and depletes cell migration in NSCLC [67]. miR-124 is a tumor suppressor in lung adenocarcinoma associated with epithelial-to-mesenchymal (EMT) phenotypes and targets the enhancer of zeste homolog 2 (*EZH2*) to suppress lung cancer cells like A549, H1299, SPC-A1, and H1975 [68]. 

### 3.5. Other Cancer Types

Cervical cancer (CC) is associated with high morbidity and mortality in women. miR-124 is downregulated in CC cell lines HeLa and SiHa [69]. Astrocyte-elevated gene-1 (*AEG-1*), an oncogene, is involved in tumor progression and chemotherapy resistance. Studies found that *AEG-1* has been upregulated in CC patients. miR-124 targets *AEG-1* and inhibits cell proliferation and migration, in addition to invasion [70]. Restoration of miR-124 suppresses the inhibitor of apoptosis-stimulating protein of *p53* (iASPP) and insulin-like growth factor 2 mRNA-binding protein 1 (*IGF2BP1*) expression and attenuates the CC cells’ growth and invasions [71]. 

In colorectal cancer (CRC), miR-124-3p.1 inhibits the CRC cells like HCT116, and suppresses cell proliferation and migration along with invasion via downregulation of AKT3 [72]. miR-124 inhibits CRC cell growth via targeting DNA methyltransferase 3B (*DNMT3B*) and *DNMT1*. miR-124 modulation increases radiosensitivity, targets paired related homeobox 1 (PRRX1), and inhibits the growth of CRC cell lines SW480 and SW620 [73]. miR-124 targets *STAT3* and represses CRC cell proliferation and growth [74]. 

Pancreatic ductal adenocarcinoma (PDAC) progression can be halted by miR-124 modulation that targets monocarboxylate transporter-1 (*MCT-1*), integrin α3 (*ITGA3*), and integrin β1 (*ITGB1*). miR-124 presents as a therapeutic biological strategy to treat PDAC [75]. Expression of miR-124 was lower in various PDAC cells like AsPC-1, PANC1, and SW1990. Exogenous transfection of miR-124 into these cells suppressed the metastatic potential and induced apoptosis [9]. miR-124 can act as a diagnostic tool in PDAC patients [12]. miR-124 targets Ras-related C3 botulinum toxin substrate 1 (Rac1) and inactivates the MKK4-JNK-c-Jun pathway. This inhibits the proliferation and invasion of pancreatic cancer cells (PCC). miR-124 inhibits *MCT1* leading to cell acidification, which represses PCC [4].

Gastric cancer is the fourth highest cause of cancer-related deaths globally. Unfortunately, patient outcomes are often unfavorable due to the recurrence of tumors and metastasis. Previous studies have shown that a rise in *HRCT1* expression is a clear indication of poor prognosis among individuals diagnosed with gastric cancer. *HRCT1* actively promotes tumor growth by activating the ERBB2-MAPK pathway. Furthermore, it has been found that *HRCT1* is negatively regulated by miR-124-3p [76]. In summary, Table 1 showcases several studies highlighting the significant impact of miR-124 on different types of cancers.

The aberrant expression of miR-124 in various cancer types suggests the multifaceted role of miR-124 as a diagnostic marker, predictor of tumor progression, and as a therapeutic target. miR-124 exerts its anti-cancer effects by acting as a tumor suppressor gene and targeting proteins (*p53*, *AKT*, *Caspase-3*, and *EZH2*), which regulates cell proliferation and other hallmarks of cancers.

## 4. miR-124: Clinical Prospects 

Cervical lesions that are considered high risk for developing into cancer are classified as either stage 2 or stage 3 cervical intraepithelial neoplasia (CIN 2 and 3). CIN 3 is particularly concerning as it is a direct precursor to invasive cancer, with a high likelihood of progression and a close correlation to the final histological diagnosis. A recent clinical study was aimed to determine the role of miR-124 and FAM19A4 on the methylation rate of CIN 2 regression, persistence, or progression in women (NCT05624827). 

Whilst clinical trials with miR-124 restoration in oncology are on the rise, there are many other studies currently being conducted using the miR-124 replacement approach. Multipotent mesenchymal stem cells (MSC)-derived exosomes induce neurovascular renovation and functional retrieval after a stroke. Circulating miR-124 were analyzed from the serum samples of post-stroke recovered patients (NCT04323501) [89]. miR-124 loaded exosomes (exo) alleviate brain injury and induce neurogenesis. Studies have been performed with exo-miR-124 delivered by stereotaxic implantation or intraparenchymal route to cerebrovascular accident (CVA) patients to prevent the disability (NCT03384433) [90]. Circulating miR-124 levels have been evaluated in coronavirus disease 2019 (COVID-19) patients with or without pneumonia and severe acute respiratory syndrome (SARS) to predict the potential benefits of this marker in Turkish populations (NCT04411563) [91]. In a randomized double-blind study to evaluate the safety and effectiveness of the ABX464 in patients with moderate to severe ulcerative colitis (NCT03760003) and/or Crohn’s disease (NCT03905109), the expression of miR-124 has been found to be increased in total blood and rectal tissues [92,93]. miR-124 level was evaluated at 3 and 6 months from baseline in the secondary outcome to assess pain sensitization in RA patients (NCT03815578) [94]. Changes in the level of miR-124 were measured in curcumin-treated familial adenomatous polyposis patients (NCT00641147) [95]. Alterations of miR-124 were monitored in ABX464 treated COVID-19 patients (NCT04393038) [96]. These studies highlight the potential for more therapeutic developments for miR-124 based approaches in various cancers given the critical roles played by miR-124. 

## 5. Hurdles and Challenges to Delivery of miR

Current challenges associated with the delivery of miR are the off-target effects that could lead to potential toxicities and reduce therapeutic efficacy. miR is targeted by the cellular host defense system. Delivery of miR-124 is limited to inflamed or infected tissues as the pathological state may alter the level of various mRNA (Figure 2). These limitations should be circumvented to avoid effects on normal surrounding tissues. Immune system activation triggered by miR can cause adverse effects to recipients [97,98]. Degradation by nucleases and deprived cell membrane penetration is another drawback associated with the delivery of miR. A lower binding affinity to the complementary sequences and undesired target tissues has been another hurdle to effective delivery. Maintaining stability and consistency is one of the challenges associated with the delivery of miR. These negative consequences can be overcome by chemical modification with oligonucleotides.

Various delivery systems are utilized for miRNA therapeutics, including lipid-based and polymer-based vectors, as well as ligand-oligonucleotide conjugate systems. These delivery methods are taken up through diverse endocytosis mechanisms. To prevent lysosomal degradation, it is crucial to facilitate endosomal escape of the RNA therapeutic. Several tactics can be employed, such as cationic lipids, viral or bacterial agents, cell-penetrating peptides (CPPs), and exosomes [97]. Of these tactics exosome mediated delivery is more effective than other delivery systems. Exosomes are tiny vesicles secreted by various cells that aid in intercellular communication by delivering different types of cargo, such as proteins (cytoplasmic proteins and membrane proteins), lipids, and nucleic acids (DNA, mRNAs, and ncRNAs) that are involved in cell-to-cell communication. These vesicles operate on a nanoscale level [99]. Recent studies have shown that exosomes derived from CT-26 are a valuable source of multiple antigens that can trigger an antitumor immune response. Additionally, these exosomes also serve as natural carriers for delivering miR-124-3p mimic. It has been discovered that TEXomiR effectively stimulates an antitumor immune response, leading to a reduction in tumor growth and an increase in survival rates [100].

## 6. Future Perspectives

Off-target effects and limited efficacy are the major drawbacks of existing therapeutic strategies such as monoclonal antibodies and several small molecule inhibitors. There are many genes involved in disease progression which cannot be directly targeted by small molecule inhibitors but can be achieved by using miR-based therapy. miR therapeutics facilitate the ability of targeting specific genes involved in the regulation of pathways involved in disease development and progression. 

Several studies have demonstrated the ability of miR-124 in the regulation of genes involved in the disease progression of various pathological conditions. However, extensive studies are required to determine the therapeutic efficacy of miR-124 for different malignancies. As well as challenges with in vivo application, some of the major obstacles to miR’s efficacy are its immune reaction and degradation; its therapeutic concentration and duration also pose a problem, and as such, novel miR delivery strategies are required. 

## 7. Conclusions

miR-124 is a promising therapeutic target in oncology due to its dysregulated expression in cancer and its roles in regulating the immune system and neurogenesis. Developing strategies to restore or suppress miR-124 expression could be a potential avenue for treating certain types of cancer. However, further research is needed to fully understand the underlying mechanisms and potential side effects before miR-124-based therapies can be developed for clinical use.

## Figures and Tables

**Figure 1 biology-12-00922-f001:**
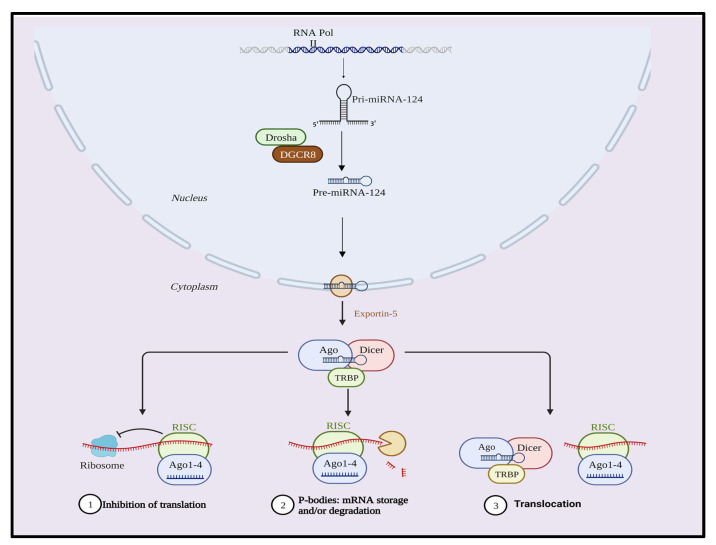
Schematic representing the biogenesis of miR-124.

**Figure 2 biology-12-00922-f002:**
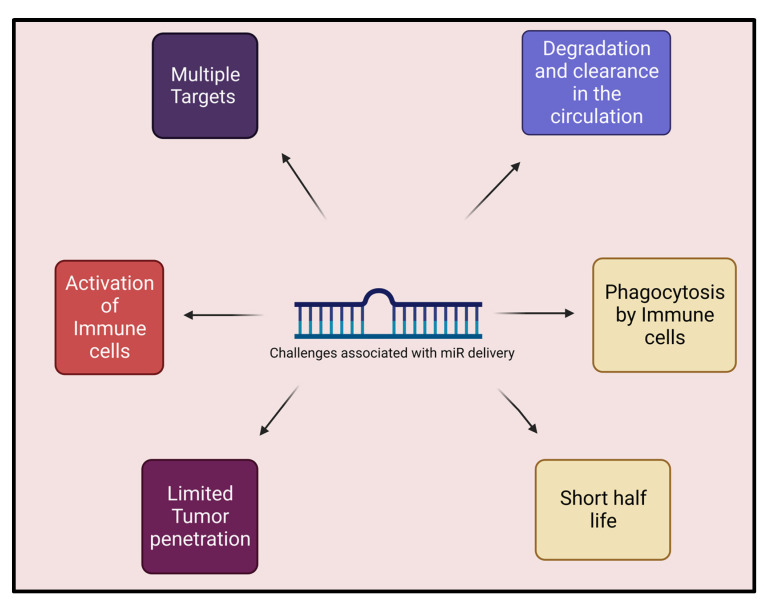
Challenges associated with microRNA delivery.

**Table 1 biology-12-00922-t001:** Summary of studies highlighting role of miR-124.

Biological Function	Disease	Subtypes	Target Gene	Regulation Pattern	Ref
**Anti-cancer activity**	Neurological cancers	Medulloblastoma	*CDK6* *NUR 77*	Regulates cell cycle progression and inhibits cell proliferation	[34,77]
		Glioblastoma	*CDK6* *SCP-1* *ROCK1* *STAT3* *MMP-9*	Regulates cell cycle progression and inhibits cell proliferation	[21,78]
	Breast cancer	Triple-negative breast cancer	*IL-11* *CD-151* *CDK-4* *EGFR*	Inhibits cell proliferation, metastasis	[42,79,80]
	Hepatocellular carcinoma		*SP1* *STAT3* *CAS3*	Increases apoptosis and reduces cell proliferation	[51]
	Lung cancer	NSCLC	*ZEB-1* *SOX-9* *CDH-2*	Prevents migration and invasion	[81,82]
	Cervical cancer	-	*AEG-1* *P53*	Induces apoptosis and decreases cell proliferation	[83,84]
	Colorectal cancer	-	*AKT-3* *DNMT3B* *DNMT-1* *STAT3*	Inhibits cell proliferation, migration, and invasion	[58,72]
	Gastric cancer	-	*NOTCH-1* *PI3/AKT/STAT3*	Inhibits cell proliferation and invasion	[85,86]
	Renal cell carcinoma	-	*STAT3/MMP-9* *MEG/P53*	Regulates apoptosis and decreases cell proliferation	[87,88]
	Pancreatic cancer	-	*MCT-1* *ITGA3* *MKK4/JNK/c-Jun*	Inhibits cell proliferation and inhibition	[75]

## Data Availability

Not applicable.

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
