# Peer review of "MicroRNA (miR)-124: A Promising Therapeutic Gateway for Oncology"

_biology, 2023, doi:10.3390/biology12070922_

Round 1

Reviewer 1 Report

The review updated the role of miR-124 as a potential target in oncology, since dysregulated expression is related to various cancer types.

1. Previous Reviews describing the role of miR-124 in oncology are present, therefore the novelty of this Review is a more recent update on miR-124 and cancer. To this aim, Authors should update the review to the last 5 years, excluding (if it is possible) earlier references thus preferring latest issues, including 2023 reports.

2. Authors should improve the quality of Graphical abstract and Figure 2 since poor clear and attracting.

The use of English is clear and understandable.

Author Response

We would like to thank the reviewer for their critical review and suggestions. We have made substantial revisons to the manuscript.

The review updated the role of miR-124 as a potential target in oncology since dysregulated expression is related to various cancer types.

Previous Reviews describing the role of miR-124 in oncology are present, therefore, the novelty of this Review is a more recent update on miR-124 and cancer. To this aim, Authors should update the review to the last 5 years, excluding (if it is possible) earlier references thus preferring latest issues, including 2023 reports.

Response: As per the reviewer’s suggestion, we have included new reports in the manuscript regarding the role of miR-124 in various types of cancer.

Xu, Y.; Liu, N.; Wei, Y.; Zhou, D.; Lin, R.; Wang, X.; Shi, B. Anticancer effects of miR-124 delivered by BM-MSC derived exosomes on cell proliferation, epithelial mesenchymal transition, and chemotherapy sensitivity of pancreatic cancer cells. Aging (Albany NY) 2020, 12, 19660-19676, doi:10.18632/aging.103997.Cha, N.; Jia, B.; He, Y.; Luan, W.; Bao, W.; Han, X.; Gao, W.; Gao, Y.;

Cha, N.; Jia, B.; et al. MicroRNA‑124 suppresses the invasion and proliferation of breast cancer cells by targeting TFAP4. Oncol Lett 2021, 21, 271, doi:10.3892/ol.2021.12532.

Sanuki, R.; Yamamura, T. Tumor Suppressive Effects of miR-124 and Its Function in Neuronal Development. Int J Mol Sci 2021, 22, doi:10.3390/ijms22115919.

Ostrom, Q.T.; Patil, N.; Cioffi, G.; Waite, K.; Kruchko, C.; Barnholtz-Sloan, J.S. CBTRUS Statistical Report: Primary Brain and Other Central Nervous System Tumors Diagnosed in the United States in 2013-2017. Neuro Oncol 2020, 22, iv1-iv96, doi:10.1093/neuonc/noaa200.

Yan, C.; Kong, X.; Gong, S.; Liu, F.; Zhao, Y. Recent advances of the regulation roles of MicroRNA in glioblastoma. Int J Clin Oncol 2020, 25, 1215-1222, doi:10.1007/s10147-020-01685-y.

Zhao, Q.; Jiang, F.; Zhuang, H.; Chu, Y.; Zhang, F.; Wang, C. MicroRNA miR-124-3p suppresses proliferation and epithelial–mesenchymal transition of hepatocellular carcinoma via ARRDC1 (arrestin domain containing 1). Bioengineered 2022, 13, 8255-8265, doi:10.1080/21655979.2022.2051686.

Dong, Z.-b.; Wu, H.-m.; He, Y.-c.; Huang, Z.-t.; Weng, Y.-h.; Li, H.; Liang, C.; Yu, W.-m.; Chen, W. MiRNA-124-3p.1 sensitizes hepatocellular carcinoma cells to sorafenib by regulating FOXO3a by targeting AKT2 and SIRT1. Cell Death & Disease 2022, 13, 35, doi:10.1038/s41419-021-04491-0.

Zhu, Q.; Zhang, Y.; Li, M.; Zhang, Y.; Zhang, H.; Chen, J.; Liu, Z.; Yuan, P.; Yang, Z.; Wang, X. MiR-124-3p impedes the metastasis of non-small cell lung cancer via extracellular exosome transport and intracellular PI3K/AKT signaling. Biomarker Research 2023, 11, 1, doi:10.1186/s40364-022-00441-w.

Rezaei, R.; Baghaei, K.; Hashemi, S.M.; Zali, M.R.; Ghanbarian, H.; Amani, D. Tumor-Derived Exosomes Enriched by miRNA-124 Promote Anti-tumor Immune Response in CT-26 Tumor-Bearing Mice. Frontiers in Medicine 2021, 8.

Authors should improve the quality of Graphical abstract and Figure 2 since poor clear and attracting.

Response:  As per your suggestion, we have made some enhancements to the manuscript. We have included an upgraded graphical abstract and Figure 2.

Reviewer 2 Report

Authors of this review article with the title “microRNA (miR) - 124: A Promising Therapeutic Gateway for Oncology” had ambition to present the knowledge in miR-124 function in relation to cancer progression and its potential for anticancer therapy. They summarized miRNA biogenesis, the results of cancer cell studies on miR-124 regulated genes and their functions. They focused on neurological, breast, hepatobiliary and lung cancers and presented also several results from cervical, colorectal and pancreatic cancer cells. In clinical application, several clinical trials in expression of miR-124 or some regulated genes in several diseases, but not in cancer patients were included, maybe besides FAP, which is considered as pre-malignant state. In Table 1. they presented miR-124 functions in cancer and neurologic disease, but without any references. Furthermore, the authors discussed the problems with miR-124 delivery in association of miR-based therapy citing only one reference from dengue study. Authors concluded that further research is needed.

According to the title and abstract of this article I have await that other diseases as cancer will be omitted and the results from aberrant miR-124 expressions in cancer patients and their associations to clinical features will be included. Regarding the molecular cancer therapy, I am sure that there are studies focusing on small molecules as miRNAs delivery to the tumor location (vectors, nanoparticles, exosomes…) that could be discussed.

In addition to inadequate conception of this manuscript, I found several problems.

·        -  Firstly, there is a rule that genes need to be written by italics for discrimination from the names of relevant proteins, which are frequently the same. Make corrections in the whole text.

·         - In the relation to miRNA function, the term “target” is correctly used for inhibition of gene expression by physical binding of miRNA to mRNA with possible protein loss consequences, thereby miRNA not targets the protein.

·         - Graphical abstract is not in consistency with the content of presented article. Liver cancer was not considered in the text and on the contrary, breast and hepatobiliary cancers were here omitted. The “key” role of presented genes regulated by miR-124 in several types of cancer were not described anywhere in the text.

·         - Figure 1: Described miRNA biogenesis is not specific for miR-124, but for miRNAs generally. Note it in the text. Point 3. “Translocation” at the figure was omitted in the text.

·             - Table 1: In the column “Target genes”, the genes (italics!!!) together with protein complexes were included. Correct it and add the relevant references.

·         - The authors used several inadequate references. Ref. 6-8 -  breast and pancreas Ca?, ref. 34 breast Ca ?,  ref. 78 – not cancer… It is a question, whether is correct to refer such publications, which have not miR-124 expression in the main message/result (in title and/or in abstract) as ref. 31, 42 or 56.

·        -  If authors used data from other studies published in very similar review article (ref. 4. – Jia, X.; Wang, X.; Guo, X.; Ji, J.; Lou, G.; Zhao, J.; Zhou, W.; Guo, M.; Zhang, M.; Li, C.; et al. MicroRNA-124: An emerging therapeutic target in cancer. Cancer Med 2019, 8, 5638-5650, doi:10.1002/cam4.2489.), they need to refer the original papers, not ref. 4.

·        -  Row 80: inaccurate formulation – “miR-124 directly targets the CLOCK gene to silence the expression and target NF-kB [16]”. Instead  that “ ….CLOCK gene to silence the expression and reduce the activation of NF-kB.

·        -  Genes and proteins are written by full names, abbreviations or by both. Correct it in the same manner.

·        -  tableure 2 : Do you mean Figure 2? Weak quality of figure.

·         - To paragraph 3: The manuscript could be more readable, when the authors try for better arrangement of the text, namely in paragraphs 3.1. – 3.4., if is it possible.

I recommend major revision including changes in the content.

Minor revision.

Author Response

We thank the reviewer for their critical insights and suggestions. We have extensively modified the manuscript based on the suggestions.

Authors of this review article with the title “microRNA (miR) - 124: A Promising Therapeutic Gateway for Oncology” had ambition to present the knowledge in miR-124 function in relation to cancer progression and its potential for anticancer therapy. They summarized miRNA biogenesis, the results of cancer cell studies on miR-124 regulated genes and their functions. They focused on neurological, breast, hepatobiliary and lung cancers and presented also several results from cervical, colorectal and pancreatic cancer cells.

In clinical application, several clinical trials in expression of miR-124 or some regulated genes in several diseases, but not in cancer patients were included, maybe besides FAP, which is considered as pre-malignant state.

Response:  We have incorporated a recent clinical study on the application of miR-124 in oncology, as suggested. There is limited available clinical trial data related to miR-124 in oncology. This information has been included in the manuscript.

In Table 1. they presented miR-124 functions in cancer and neurologic disease, but without any references.

Response: Table 1 in section 3 presents a comprehensive summary of the biological functions of miR-124, supported by relevant references in the text and have now also been added to the table.

Furthermore, the authors discussed the problems with miR-124 delivery in association of miR-based therapy citing only one reference from dengue study. Authors concluded that further research is needed. According to the title and abstract of this article I have await that other diseases as cancer will be omitted and the results from aberrant miR-124 expressions in cancer patients and their associations to clinical features will be included. Regarding the molecular cancer therapy, I am sure that there are studies focusing on small molecules as miRNAs delivery to the tumor location (vectors, nanoparticles, exosomes…) that could be discussed.

Response: We thank the reviewer for this comment. As suggested, relevant references have been included in the revision and discussed delivery of miR-124.

In addition to inadequate conception of this manuscript, I found several problems.  Firstly, there is a rule that genes need to be written by italics for discrimination from the names of relevant proteins, which are frequently the same. Make corrections in the whole text.   In the relation to miRNA function, the term “target” is correctly used for inhibition of gene expression by physical binding of miRNA to mRNA with possible protein loss consequences, thereby miRNA not targets the protein - Graphical abstract is not in consistency with the content of presented article. Liver cancer was not considered in the text and on the contrary, breast and hepatobiliary cancers were here omitted. The “key” role of presented genes regulated by miR-124 in several types of cancer were not described anywhere in the text.

Response: We have taken the suggestion and amended formatting of genes and added additional information in the revised manuscript.

Described miRNA biogenesis is not specific for miR-124, but for miRNAs generally. Note it in the text. Point 3. “Translocation” at the figure was omitted in the text.  - Table 1: In the column “Target genes”, the genes (italics!!!) together with protein complexes were included. Correct it and add the relevant references.

Response: We have made corrections across the manuscript as suggested.

The authors used several inadequate references. Ref. 6-8 -  breast and pancreas Ca?, ref. 34 breast Ca ?,  ref. 78 – not cancer… It is a question, whether is correct to refer such publications, which have not miR-124 expression in the main message/result (in title and/or in abstract) as ref. 31, 42 or 56.

Response:  We have corrected relevant references and incorporated new studies to enhance the quality of the manuscript.

If authors used data from other studies published in very similar review article (ref. 4. – Jia, X.; Wang, X.; Guo, X.; Ji, J.; Lou, G.; Zhao, J.; Zhou, W.; Guo, M.; Zhang, M.; Li, C.; et al. MicroRNA-124: An emerging therapeutic target in cancer. Cancer Med 2019, 8, 5638-5650, doi:10.1002/cam4.2489.), they need to refer the original papers, not ref. 4.

Response: We have added additional references as suggested by the reviewer.

Row 80: inaccurate formulation – “miR-124 directly targets the CLOCK gene to silence the expression and target NF-kB [16]”. Instead  that “ ….CLOCK gene to silence the expression and reduce the activation of NF-kB.

Response: We have modified the statement which now reads: “GBM cells like U87MG and T98G have high expression of the CLOCK gene, which plays a vital role in maintaining tumorigenesis. miR-124 can effectively silence the CLOCK gene directly, by inhibiting the activation of NF-kB”.

Genes and proteins are written by full names, abbreviations or by both. Correct it in the same manner.

Response: We have made corrections across the manuscript as suggested.

tableure 2 : Do you mean Figure 2? Weak quality of figure.

Response:  We were made aware by the editor of a glitch that affected the quality of Figure 2 which has now been corrected.

To paragraph 3: The manuscript could be more readable, when the authors try for better arrangement of the text, namely in paragraphs 3.1. – 3.4., if is it possible.

Response: The paragraphs were rearranged and modified in sections 3.1 to 3.4, as recommended.

Reviewer 3 Report

A review by Gourishetti et al. "microRNA (miR) - 124: A Promising Therapeutic Gateway for Oncology" summarizes knowledge of the biological role of miR-124 in the pathogenesis of various human tumors.

There are several Major Comments to this review:

1. When writing the review, the authors used articles published relatively long ago. In particular, there is not a single reference to articles from 2022-2023, although such publications are available, including in open access journals.

2. Authors should edit and check the correctness of the English language in chapter 3.3 Hepatobiliary Cancer. Perhaps some sentences are better divided. In addition, it is necessary to correct the verb tense agreement.

Minor Comments:

1. Figure 2 needs to be made in a higher resolution, the inscriptions are almost unreadable.

2. Line 273. It seems to me that "Figure 2" was meant, not "tableure 2". Please, check.

It is necessary to correct the verb tense agreement.

Some sentences are too long.

Author Response

We thank the reviewer for their valuable time in critically reviewing the manuscript. Based on their suggestions, we have substantially modified the manuscript.

When writing the review, the authors used articles published relatively long ago. In particular, there is not a single reference to articles from 2022-2023, although such publications are available, including in open access journals.

Response: Thank you for the suggestion and we have now added new references.

Xu, Y.; Liu, N.; Wei, Y.; Zhou, D.; Lin, R.; Wang, X.; Shi, B. Anticancer effects of miR-124 delivered by BM-MSC derived exosomes on cell proliferation, epithelial mesenchymal transition, and chemotherapy sensitivity of pancreatic cancer cells. Aging (Albany NY) 2020, 12, 19660-19676, doi:10.18632/aging.103997.Cha, N.; Jia, B.; He, Y.; Luan, W.; Bao, W.; Han, X.; Gao, W.; Gao, Y.;

Cha, N.; Jia, B.; et al. MicroRNA‑124 suppresses the invasion and proliferation of breast cancer cells by targeting TFAP4. Oncol Lett 2021, 21, 271, doi:10.3892/ol.2021.12532.

Sanuki, R.; Yamamura, T. Tumor Suppressive Effects of miR-124 and Its Function in Neuronal Development. Int J Mol Sci 2021, 22, doi:10.3390/ijms22115919.

Ostrom, Q.T.; Patil, N.; Cioffi, G.; Waite, K.; Kruchko, C.; Barnholtz-Sloan, J.S. CBTRUS Statistical Report: Primary Brain and Other Central Nervous System Tumors Diagnosed in the United States in 2013-2017. Neuro Oncol 2020, 22, iv1-iv96, doi:10.1093/neuonc/noaa200.

Yan, C.; Kong, X.; Gong, S.; Liu, F.; Zhao, Y. Recent advances of the regulation roles of MicroRNA in glioblastoma. Int J Clin Oncol 2020, 25, 1215-1222, doi:10.1007/s10147-020-01685-y.

Zhao, Q.; Jiang, F.; Zhuang, H.; Chu, Y.; Zhang, F.; Wang, C. MicroRNA miR-124-3p suppresses proliferation and epithelial–mesenchymal transition of hepatocellular carcinoma via ARRDC1 (arrestin domain containing 1). Bioengineered 2022, 13, 8255-8265, doi:10.1080/21655979.2022.2051686.

Dong, Z.-b.; Wu, H.-m.; He, Y.-c.; Huang, Z.-t.; Weng, Y.-h.; Li, H.; Liang, C.; Yu, W.-m.; Chen, W. MiRNA-124-3p.1 sensitizes hepatocellular carcinoma cells to sorafenib by regulating FOXO3a by targeting AKT2 and SIRT1. Cell Death & Disease 2022, 13, 35, doi:10.1038/s41419-021-04491-0.

Zhu, Q.; Zhang, Y.; Li, M.; Zhang, Y.; Zhang, H.; Chen, J.; Liu, Z.; Yuan, P.; Yang, Z.; Wang, X. MiR-124-3p impedes the metastasis of non-small cell lung cancer via extracellular exosome transport and intracellular PI3K/AKT signaling. Biomarker Research 2023, 11, 1, doi:10.1186/s40364-022-00441-w.

Rezaei, R.; Baghaei, K.; Hashemi, S.M.; Zali, M.R.; Ghanbarian, H.; Amani, D. Tumor-Derived Exosomes Enriched by miRNA-124 Promote Anti-tumor Immune Response in CT-26 Tumor-Bearing Mice. Frontiers in Medicine 2021, 8

Authors should edit and check the correctness of the English language in chapter 3.3 Hepatobiliary Cancer. Perhaps some sentences are better divided. In addition, it is necessary to correct the verb tense agreement.

Response: We have edited the language in section 3.3 as suggested.

Minor Comments:

Figure 2 needs to be made in a higher resolution, the inscriptions are almost unreadable.

Response: We were made aware by the editor of a glitch that affected the quality of Figure 2 which has now been corrected.

Line 273. It seems to me that "Figure 2" was meant, not "tableure 2". Please, check.

Response: This error has been corrected as suggested.

Round 2

Reviewer 1 Report

The Authors improved figures and the whole manuscript with new references and contents. I recommend publication in Biology journal.

Author Response

We thank the reviewer for their time and positive intent for publication.

Reviewer 2 Report

Authors made many changes for improving this manuscript, but I have several recommendations yet.

-       They do not meet reviewer recommendation “Figure 1: “Translocation” at the figure was omitted in the text.

-       Paragraph 3.3: Are the hepatobiliary cancers the only among HCC, which have the high mortality rate? This paragraph is mostly about HCC.

-       Paragraph 3.5: Gastric cancer could be added before “The aberrant expression of miR-124 in various cancer types….”

-       Paragraph 4: Whether only one clinical trial in cancer patients regarding to miR-124 clinical utility exist, it can be noticed in Future perspectives or Conclusions. Delete this paragraph, that means   any miR-124 clinical implications in other human diseases in this article titled “microRNA (miR) - 124: A Promising Therapeutic Gateway for Oncology”

-       Paragraph 5: Challenges to miR-124 delivery hold for all miRNAs, not specifically for miR-124.

-       Paragraph 6: Similarly, focus only on cancer.

-       The manuscript contents mix of incorrect italic and normal fonts in the names of genes and proteins (protein complexes), yet. miRNAs target genes not proteins; therefore, for example “ miRNAxxx down-regulate  zzz protein through the targeting of its encoding gene”. For example PI3K/Akt/NFkB signaling pathway is protein complex (do not write in italics). Make corrections carefully.

-       Table 1: The references need to be at extra column, not in target genes. Are all members of four protein complexes targeted by miR-124?

Minor revision.

Author Response

We thank the reviewer for their time and continued efforts to enhance the quality of the manuscript. Please see point-by-point response for this round below:

Authors made many changes for improving this manuscript, but I have several recommendations yet.

  • They do not meet reviewer recommendation “Figure 1: “Translocation” at the figure was omitted in the text.

Response: In the section discussing miRNA biogenesis, the importance of miR translocation is now discussed.

  • Paragraph 3.3: Are hepatobiliary cancers the only among HCC, which have the high mortality rate? This paragraph is mostly about HCC.

Response: The section has been modified based on the suggestion.

  • Paragraph 3.5: Gastric cancer could be added before “The aberrant expression of miR-124 in various cancer types….”

Response: Following the suggestion, the sentences were relocated in the revised version of the manuscript.

  • Paragraph 4: Whether only one clinical trial in cancer patients regarding to miR-124 clinical utility exists, it can be noticed in Future perspectives or Conclusions. Delete this paragraph, that means   any miR-124 clinical implications in other human diseases in this article titled “microRNA (miR) - 124: A Promising Therapeutic Gateway for Oncology”

Response: We thank the reviewer for this comment. Keeping this in mind, we have renamed the section to Clinical Prospects to highlight the need for development of for miR-124 therapeutics in oncology keeping in mind the other studies that are on-going to provide readers a point of comparison.

  • Paragraph 5: Challenges to miR-124 delivery hold for all miRNAs, not specifically for miR-124

Response: As suggested, the paragraph has been modified to address the challenges related to miR delivery.

  • Paragraph 6: Similarly, focus only on cancer.

Response: As suggested, other disorders have been removed from the section.   

  • The manuscript contents mix of incorrect italic and normal fonts in the names of genes and proteins (protein complexes), yet. miRNAs target genes not proteins; therefore, for example “ miRNAxxx down-regulate  zzz protein through the targeting of its encoding gene”. For example PI3K/Akt/NFkB signaling pathway is protein complex (do not write in italics). Make corrections carefully.

Response: We have now made corrections as suggested.

  • Table 1: The references need to be at extra column, not in target genes. Are all members of four protein complexes targeted by miR-124?

Response: A separate column has been added for references in the table, as suggested., and previous reports suggest that miR-124 acts on multiple targets

Reviewer 3 Report

I thank the authors for the work done to correct the shortcomings. In this form, the review can be accepted for publication.

Author Response

(The authors gave the same response as above.)
